# No evidence for increased fitness of offspring from multigenerational effects of parental size or natal carcass size in the burying beetle *Nicrophorus marginatus*

Ethan P. Damron[◔], Ashlee N. Smith[‡], Dane Jo[◍][‡], Mark C. Belk[◍]*[◔]

Department of Biology, Brigham Young University, Provo, Utah, United States of America

◔ These authors contributed equally to this work.
‡ These authors also contributed equally to this work.
* mark_belk@byu.edu

**Data Availability Statement:** All the data used in this study are available from the Dryad database (DOI: https://doi.org/10.5061/dryad.1vhhmgqt0).

## Abstract

Multigenerational effects (often called maternal effects) are components of the offspring phenotype that result from the parental phenotype and the parental environment as opposed to heritable genetic effects. Multigenerational effects are widespread in nature and are often studied because of their potentially important effects on offspring traits. Although multigenerational effects are commonly observed, few studies have addressed whether they affect offspring fitness. In this study we assess the effect of potential multigenerational effects of parental body size and natal carcass size on lifetime fitness in the burying beetle, *Nicrophorus marginatus* (Coleoptera; Silphidae). Lifespan, total number of offspring, and number of offspring in the first reproductive bout were not significantly related to parental body size or natal carcass size. However, current carcass size used for reproduction was a significant predictor for lifetime number of offspring and number of offspring in the first brood. We find no evidence that multigenerational effects from larger parents or larger natal carcasses contribute to increased fitness of offspring.

## Introduction

Multigenerational effects (often called maternal effects) are components of the offspring phenotype that are the result of the parental phenotype and the parental environment as opposed to heritable genetic effects [1–4]. These effects are widespread in nature and have been extensively studied in invertebrates [3, 5–8]. Multigenerational effects have been reported in response to such factors as nutrients, egg size, the presence of predators, maternal age, maternal body size, and disease resistance [6, 9–12]. The phenotypic changes that occur in offspring as a result of multigenerational effects include, changes in body size, age at maturity, developmental rate, dispersal behavior, survival, lifespan, and diapause [13–22]. Such traits are often assumed to contribute to evolutionary fitness of offspring, but tests of this connection are few.

One of the most frequently reported multigenerational effects is variation in body size. Body size is well studied because of its strong, positive relationship with fitness in some species

**Funding:** The authors received no specific funding for this work.

**Competing interests:** The authors have declared that no competing interests exist.

[23–25], and because an individual's size is typically easy to measure. Parental body size has been shown to cause multigenerational effects in offspring where larger females lay larger eggs [11, 26–32], which then hatch into larger offspring. Natal environment can also cause multigenerational effects on body size. On poor quality food resources, females tend to lay fewer, but larger eggs [9, 33, 34]. There is also a positive correlation between resource availability and offspring size [35, 36]. In crowded environments with more competition for resources, females tend to lay fewer, larger eggs, which theoretically gives offspring a size advantage when competing for resources [37–40]. Although there is ample evidence for multigenerational effects on offspring body size, it is unclear whether those differences in body size of offspring translate into increased fitness of offspring.

The overall fitness of an individual can be measured as the total number of offspring produced over a lifetime [41, 42]. Estimating fitness in natural populations is difficult [43–46], and many researchers use phenotypic traits such as body size, ejaculate size, and egg load as predictors of lifetime fitness [35, 47–49]. However, the link between these measures and lifetime fitness is often assumed, but not usually empirically tested (i.e., [50–52]). Increased offspring body size resulting from multigenerational effects does not necessarily lead to an increase in fitness of offspring [4, 53]. Hence, the need to examine the relationship between multigenerational effects and resulting fitness empirically.

Burying beetles (genus *Nicrophorus*) are ideal organisms for studies of multigenerational effects of body size. These beetles use small vertebrate carcasses for food and reproduction, and the carcass serves as the sole food resource for both parents and offspring during reproduction [33], thus effects of parental size and natal carcass size can be disentangled. The parents provide their offspring extensive parental care in the form of regurgitating predigested carrion and defending the larvae from intruders [54, 55]. Adult body size is determined by the amount of carrion that an individual consumes as a larva, and parents cull the brood through filial cannibalism so that the brood size matches the size of the carcass [33, 56], resulting in a positive correlation between offspring number and carcass size [33, 39]. In at least one species, parents also cull the brood according to their competitive environment so that they raise fewer, larger offspring when competition for carcasses is high, and more, smaller offspring when competition is low [39]. Body size is important for competitive interactions because inter- and intraspecific competition for carcasses can be intense [57–60], and body size generally determines the winners of competitions in both males and females [35, 57, 61–65]. However, the importance of body size to fitness is poorly known in most species of burying beetles (see [33, 65]).

Previous studies in burying beetles have demonstrated multigenerational effects on offspring body size. For example, larger offspring being produced on larger carcasses [33, 65], larger mothers laying larger eggs, which hatch into larger offspring [11], young mothers producing smaller offspring [66], mothers producing smaller offspring when the male parent is present [52], mothers altering hormone concentration in eggs [67], and mothers changing egg laying and offspring care behaviors when deprived of food [68]. Body size of the parent and carcass size are dominant mechanisms through which multigenerational effects are transmitted in burying beetles. Parents control the size of offspring on a carcass in two ways. First, by culling first instar larvae parents that leave fewer offspring on a carcass will produce larger offspring [56]. Second, parents consume a certain portion of the carcass for themselves, thus parents that consume less of the carcass can produce larger offspring on a given carcass size [33, 39]. Larger carcasses provide more resources for both parents and their offspring, thus larger carcasses often result in larger offspring [33]. Clearly, multigenerational effects on offspring size in burying beetles may be transmitted through both parental size and carcass size (i.e., natal environment). The implied prediction is that these multigenerational effects are

anticipatory and adaptive such that larger offspring would produce more offspring over their lifetime and thus express increased fitness [4]. However, multigenerational effects may not be adaptive for offspring, rather they can evolve only to increase fitness of the parents [4]. Whether multigenerational effects on body size in burying beetles are adaptive for offspring and translate into increased fitness of offspring is not known.

In this study we tested whether multigenerational effects of parental size and natal carcass size that result in increased offspring size in the burying beetle *Nicrophorus marginatus* [65] also cause an increase in lifetime fitness of offspring (i.e., they are adaptive [4]). We use a two-generation experimental design combined with a measure of lifetime fitness to assess possible fitness effects transmitted through parent body size or quality of natal environment.

## Methods

### Source of burying beetles

To generate the laboratory-bred population to use for the experiment, we captured adult *N. marginatus* at the Utah Wetland Preserve near Goshen, Utah, USA (this is public land, and no permits were required to trap or collect beetles), in August 2014 using pitfall traps baited with aged chicken. We created 41 independent genetic lines from wild-caught pairs by providing a 40g mouse carcass for each pair and allowing them to breed. We designated the offspring from the wild-caught pairs as the first parental generation. After eclosion and emergence of this first laboratory-bred generation, we placed individuals in small plastic containers (15.6 x 11.6 x 6.7 cm), fed them *ad libitum* raw chicken liver twice weekly, and maintained them on a 14:10 h light:dark cycle at 21°C until they were used in experiments. We designated the date of eclosion as the first day of life for all beetles used in the experiments.

### Experimental design

The purpose of this experiment was to determine the effect on lifetime fitness of potential multigenerational effects of parental body size and natal carcass size in *N. marginatus*. We measured three fitness response variables: lifespan, total number of offspring, and number of offspring produced in the first brood. Number of offspring in the first brood was used in addition to lifetime number of offspring because number in the first brood may or may not follow total lifetime numbers [39], and in the natural environment, number of reproductive bouts may be fewer than that observed in laboratory experiments. Thus, number in the first brood may represent natural conditions better than total lifetime number of offspring in a laboratory environment. To generate potential multigenerational effects from parental body size and natal carcass size, we allowed large and small beetles to reproduce on large or small carcasses (first parental generation). To assess fitness effects of multigenerational effects, we then allowed female offspring from this first parental generation to reproduce throughout their lifetime. We then measured the three fitness responses for all females of this second generation and compared them among treatment combinations.

To determine what sizes constituted large or small beetles for assignment in the first parental generation, we used the distribution of pronotum widths from wild-caught beetles. We assigned beetles from the first parental generation with pronotum widths > 1 standard deviation above and < 1 standard deviation below the mean (derived from wild-caught population) as large and small, respectively. In the wild-caught population, the mean pronotum width of females was 6.67mm, with a standard deviation of 0.78mm (N = 50). The mean pronotum width of males was 6.87mm, with a standard deviation of 0.72mm (N = 50). Thus, the size range of large and small female beetles that we used in this experiment for the first parental generation was 7.44mm– 8.22mm and 5.11mm– 5.89mm, respectively, and the corresponding

size range for large and small males was 7.60mm– 8.32mm and 5.42mm– 6.15mm, respectively. For this first part of the experiment, we used a fully crossed factorial design. There were four parental size treatments—large male with large female, large male with small female, small male with large female, and small male with small female. Each parental size treatment was crossed with both small (20g) and large (40g) carcass sizes, for a total of eight treatment combinations. We chose 20g and 40g carcass sizes based on a previous study that tested multiple carcass sizes from 5g to 50 g. The 20g and 40g sizes were both within the range of carcass sizes whereon *N. marginatus* experienced equally high reproductive success [69]. We completed six replicates of each of the eight treatment combinations resulting in an initial sample size of 48 pairs for the first parental generation.

We began each first-generation experimental replicate by randomly choosing a genetically unrelated pair of beetles that fit into one of the parental size treatments. We placed the pair in a small brood container and randomly assigned them either a 20g (± 1.0g) or a 40g (± 2.0g) mouse carcass. We checked the brood containers daily for larvae, and after larvae arrived on the carcass, we removed the lid of the small brood container and placed the small container in an abandonment chamber (see [70] for details). We checked abandonment chambers daily for leaving adults or dispersal of larvae, and when the carcass was consumed and larvae dispersed into the soil, we removed and weighed the remaining parent(s). The larvae from each brood reached eclosion 4–5 weeks after dispersal. We weighed each newly-eclosed adult offspring, placed them in an individual container, and fed them *ad libitum* chicken liver until they reached sexual maturity. We used results from this first-generation experiment to test for effects of body size of parents (male and female separately) and carcass size on reproductive output and offspring traits [65]. In the first-generation experiment female body size generated significant effects on offspring body size, but male size had no effect on offspring traits and there were no significant interactions between male and female body size [65]. For this reason, we included parental size treatments as four independent treatments in analysis of the second-generation experiment (this paper) rather than as a factorial. This reduced complexity of the second-generation analysis by eliminating some two-way and three-way, and all four-way interactions. For additional information on the first parental generation methods and results from the experiment, see Smith and Belk [65].

For the second parental generation, we used the offspring from the first parental experiment described above to determine how the size of parental *N. marginatus* beetles and the size of carcass that they reproduced on affected lifetime fitness of their offspring. For the second parental generation, we randomly chose two female offspring from each of the first parental replicates for the experiment. These two females (from replicates of the eight first generation treatment combinations) were randomly assigned either large (40g) or small (20g) carcasses for reproduction, for a total of sixteen treatment combinations in the second experiment. We started six replicates for each of the sixteen treatments for a total of 96 replicates. Five experimental pairs failed to reproduce (one each for five of the treatment combinations), so the realized sample size was 91 pairs.

To determine fitness of second-generation beetles, sexually mature females (age > 21 days) were paired with a randomly selected, sexually mature, genetically unrelated male, and the pair was placed in a plastic container (14 × 13 × 17cm) filled with 10cm of moist soil and given either a 20g (±1.0g) or a 40g (± 2.0g) mouse carcass (based on the random assignment described above). The containers were kept in an environmental chamber at 21°C on a 14:10 h light:dark cycle. We checked for larvae daily, and after larvae arrived on the carcass, we removed the male from the container so that only the female cared for the offspring. After the larvae dispersed into the soil to pupate, we removed the female and placed her in a small container with *ad libitum* chicken liver for two days, and then set her up to mate again with a new,

randomly selected, genetically unrelated, virgin male on the same size carcass (20g or 40g). This cycle continued for her entire life. In the second-generation experiment, number of successful reproductive bouts ranged from 1 to 5 with a mean of 2.9 bouts. We measured lifespan as the number of days the female survived after eclosion. After the offspring from each brood of this second-generation experiment enclosed as adults, we recorded the sex, weight, and pronotum width for each beetle. We summed the number of offspring from each brood to determine a female's lifetime number of offspring.

## Analysis

To determine if parental size or natal carcass size influences fitness of offspring we used a mixed model analysis of covariance (ANCOVA) approach (SAS, Proc MIXED). We used three response variables to represent fitness of females from the second parental generation: i.e., total lifetime number of offspring, total number of offspring in the first brood, and lifespan. Predictor variables for each of the three models were first parental generation body size (four levels), natal carcass size (two levels), current carcass size (two levels), and pronotum size of second parental generation focal female (covariate). Initially we included all two-way and the three-way interactions of main effects in the model. However, interaction terms were not significant in any of the models, so, we report results from a reduced model that included only one, two-way interaction–natal carcass size by current carcass size. This interaction was of interest because of the significant effect of current carcass size on fitness measures, and the possibility of a multigenerational effect that natal carcass size might provide some "priming" for efficiency of use of a similar carcass size. Because we used two individuals from each of the first-generation broods to breed for the second generation, we used first generation parental brood ID as a random effect to account for the relatedness of individuals from the same brood. We used raw data for the analysis, and inspection of residual plots showed no departure from assumptions for the parametric model.

## Results

None of our three measures of fitness were significantly related to parental body size (first generation parents) or natal carcass size. Current carcass size was a significant predictor for lifetime number of offspring, and number of offspring in the first brood. Pronotum width of the female was a significant predictor of life span of the female (Table 1). On average lifetime number of offspring was about 20% less on small carcasses compared to large carcasses; and the number of offspring in the first brood was about 28% less on small carcasses compared to large carcasses (Table 2). The largest females (7.7 mm pronotum width) lived up to 19 days longer than the smallest females (5 mm pronotum width), but this increased lifespan had no effect on lifetime number of offspring (Table 1).

## Discussion

Multigenerational effects have been widely studied and reported, especially in invertebrates [3, 5–8]. Despite ample evidence of multigenerational effects, their relationship with offspring fitness is not well established. In *N. marginatus* burying beetles, we previously documented two mechanisms that resulted in multigenerational effects on offspring body size–larger females produced larger offspring, and larger reproductive carcasses resulted in larger offspring [65]. However, larger body size in response to larger natal carcass size or larger parent size did not result in increased fitness measured as lifetime number of offspring. In many other species, body size is considered a strong predictor of fitness, and body size variation through maternal effects can lead to large differences in individual fitness [23–25]. Why do multigenerational

**Table 1. Results of mixed model analysis of covariance for three fitness measures for *N. marginatus*.**

| Dependent Variable | Effect | Num df/Den df | F-Value | p-value |
|---|---|---|---|---|
| *Lifetime Number of Offspring* | Parental Size | 3/35.7 | 0.25 | 0.8637 |
| | Natal Carcass Size | 1/35 | 0.33 | 0.5676 |
| | Carcass Size | 1/48.8 | 6.26 | **0.0158** |
| | Pronotum | 1/82.4 | 2.49 | 0.1185 |
| | Natal Carcass Size*Carcass Size | 1/48.4 | 0.04 | 0.8455 |
| *Lifespan of Female* | Parental Size | 3/39.2 | 0.77 | 0.5190 |
| | Natal Carcass Size | 1/38.4 | 1.08 | 0.3056 |
| | Carcass Size | 1/53.3 | 0.04 | 0.8483 |
| | Pronotum | 1/81.8 | 9.92 | **0.0023** |
| | Natal Carcass Size*Carcass Size | 1/52.8 | 1.45 | 0.2343 |
| *Number of Offspring from First Bout* | Parental Size | 3/27.4 | 1.65 | 0.2011 |
| | Natal Carcass Size | 1/26.7 | 0.45 | 0.5082 |
| | Carcass Size | 1/44 | 13.36 | **0.0007** |
| | Pronotum | 1/77.6 | 3.41 | 0.0687 |
| | Natal Carcass Size*Carcass Size | 1/43.5 | 0.01 | 0.9385 |

effects of increased body size have no effect on lifetime fitness in *N. marginatus*? The strongest determinant of fitness in our study was size of carcass available for reproduction. Females, of any size, producing offspring on larger carcasses, produced more offspring over a lifetime than those producing offspring on smaller carcasses. These results contrast with those from *N. vespilloides* where larger females had a reproductive advantage (i.e., produced more offspring) on

**Table 2. Least squares means and upper and lower confidence intervals for main effects of current carcass size and the interaction between current carcass size and natal carcass size in *N. marginatus*.**

| Effect | Mean | Lower CI (95%) | Upper CI (95%) |
|---|---|---|---|
| Lifetime Number of Offspring | | | |
| Carcass (20 g) | 40.74 | 34.69 | 46.79 |
| Carcass (40 g) | 50.79 | 44.72 | 56.86 |
| Natal Carcass (20 g)*Carcass (20 g) | 39.02 | 30.20 | 47.85 |
| Natal Carcass (20 g)*Carcass (40 g) | 49.86 | 41.69 | 58.03 |
| Natal Carcass (40 g)*Carcass (20 g) | 42.45 | 34.17 | 50.73 |
| Natal Carcass (40 g)*Carcass (40 g) | 51.72 | 42.75 | 60.68 |
| Lifespan of Female (days) | | | |
| Carcass (20 g) | 60.89 | 57.38 | 64.40 |
| Carcass (40 g) | 60.43 | 56.90 | 63.96 |
| Natal Carcass (20 g)*Carcass (20 g) | 60.83 | 55.71 | 65.95 |
| Natal Carcass (20 g)*Carcass (40 g) | 63.22 | 58.46 | 67.98 |
| Natal Carcass (40 g)*Carcass (20 g) | 60.95 | 56.12 | 65.77 |
| Natal Carcass (40 g)*Carcass (40 g) | 57.64 | 52.44 | 62.85 |
| Number of Offspring from First Bout | | | |
| Carcass (20 g) | 14.03 | 11.94 | 16.13 |
| Carcass (40 g) | 19.36 | 17.26 | 21.46 |
| Natal Carcass (20 g)*Carcass (20 g) | 13.46 | 10.40 | 16.53 |
| Natal Carcass (20 g)*Carcass (40 g) | 18.90 | 16.07 | 21.73 |
| Natal Carcass (40 g)*Carcass (20 g) | 14.60 | 11.73 | 17.47 |
| Natal Carcass (40 g)*Carcass (40 g) | 19.81 | 16.70 | 22.93 |

larger carcasses and smaller females had an advantage on smaller carcasses [71]. This difference observed between our study and that of Hopwood et al. [71] may be a reflection of differences in individual ecology between species [65]. Additional studies of the relationship between body size and carcass size and the consequences for reproductive output and lifetime fitness (e.g., [69, 72]) would help to determine the generality of our results.

Another interesting point from our data is that lifespan was not influenced by size of carcass used for reproduction, but lifetime number of offspring was significantly influenced by carcass size. Although we would expect some relationship between fitness and lifespan in general, across the range of lifespans observed in this experiment there was no relationship. In a study on the burying beetle *N. orbicollis*, carcass size significantly influenced lifespan, but fitness (measured as lifetime number of offspring) did not differ between larger and smaller carcasses [72]. This was interpreted as different reproductive strategies resulting in equal fitness–females reproducing on larger carcasses produced more offspring in each reproductive bout, but had shorter lifespans and thus fewer reproductive bouts; whereas, females reproducing on smaller carcasses produced fewer offspring per reproductive bout, but had longer lifespans and more reproductive bouts [72]. Our results are not consistent with this pattern or interpretation, but rather suggest that in *N. marginatus* larger carcasses produce greater lifetime number of offspring compared to smaller carcasses. This difference between studies may reflect variation in the relationship between carcass size and fitness between species of burying beetles. Generally, optimal carcass size scales with body size in burying beetles; however, relationships between carcass size and fitness are not well known in most species of burying beetles. Variation in the efficiency of use of various carcass sizes has been documented among a few species [69, 72]. Additional studies that compare lifetime reproductive output on various carcass sizes would help determine general patterns of variation in reproductive strategies among species and carcass sizes.

Multigenerational effects in burying beetles (genus: *Nicrophorus)* are well established [11, 33, 52, 65–68]. Most of these studies investigate the relationship between maternal size and offspring size [11, 33, 52, 65, 66, 73]. However, none of these studies investigated whether size related multigenerational effects affect offspring lifetime fitness. Two studies investigated the relationship between multigenerational effects and offspring survival but did not study their role in lifetime fitness of offspring [66, 68], and our data suggest that longer lifespan does not necessarily contribute to increased number of offspring over a lifetime (i.e., fitness). In natural environments, number of successful reproductive bouts may be fewer than that in our non-competitive lab environment. For this reason, we included number of offspring in the first bout as a potentially more natural alternative to measure lifetime reproductive success. Patterns were the same whether we measured fitness as lifetime number of offspring or number of offspring in the first bout, suggesting that even if females reproduce on only one carcass during their lifetime, there is still no effect of parental body size or natal carcass size.

The lack of a multigenerational effect of body size on offspring fitness that we found in this study may be due to the absence of competition in the lab environment. In our experiment, we allowed female offspring in the second generation to reproduce regardless of their size, which might not happen in nature. Burying beetles compete for access to carcasses, and generally, the largest male and female beetles dominate the carcass for reproduction [57, 60, 65]. In our study, the only factor that affected fitness was the size of the carcass on which the female reproduced, with more offspring produced on larger carcasses (Table 1). Other studies have shown that larger females have larger offspring [65, 73], which could confer a fitness advantage of large size through a maternal effect under more natural conditions where size determines whether or not a female wins access to a carcass and how many carcasses she dominates. Thus, large body size of a female beetle may translate into increased lifetime fitness in her offspring

in a natural environment because larger female offspring are able to win access to more and larger carcasses, and thus produce more offspring.

We acknowledge that competitive ability linked to individual body size may confer a fitness advantage under some circumstances, especially if access to carcasses for reproduction is frequently contested, and if body size is often linked to success in obtaining control of carcasses. However, at least three lines of evidence suggest that competition for carcasses may not be a consistent determinant of lifetime fitness. First, competition for carcasses is dependent on the density of beetles and the availability of small vertebrate carcasses at both large and small scales. Densities of beetles and small vertebrate carcasses fluctuate dramatically among years [74, 75]. Similarly, when we trap beetles from the wild for this and other studies, we find variation in the number of beetles captured in a given trap, and a range of sizes of both males and females. This spatial variation in number and body size and the temporal variation in beetle numbers and carcass numbers among and within years suggests that competition for carcasses is likely a local phenomenon. All beetles in a given area are not present and competing for every carcass. Although large beetles may win at carcasses where they compete, some carcasses will have many competitors, and some will have few. Also, when beetles breed on a carcass, they are effectively removed from the potentially competing population for at least two weeks. Thus, the competitive environment is best represented by a mosaic of local carcass availability and local beetle distribution and availability. The outcome is a range of opportunities for beetles of many sizes to be successful at finding and breeding on a carcass. Second, burying beetles exhibit alternative reproductive strategies such that fitness can be obtained by males and females acting as satellites at the carcass burial site or mating prior to finding a carcass [76–78]. If only large individuals can gain access to and reproduce successfully on carcasses, then alternative strategies would not evolve, and body size would be more strongly constrained by selection. Alternative reproductive strategies and a range of body sizes in natural populations suggest that being large is not the only way to be successful in competition for carcasses. Third, in a previous study, we measured competitive ability to gain a carcass based on size in both male and female burying beetles [65]. The relationship differed by sex such that for females, the size difference between competitors had to be much larger for body size to be an important determinant of competitive outcomes. Even for males, there was some non-zero probability of the smaller competitor winning the carcass, except at the largest size difference [65]. Other studies on burying beetles have reported corroborating effects of beetle sizes found on carcasses in natural systems [33]. Body size may be a strong determinant of success in competing for carcasses, but in many contests, priority effects and stochasticity or "luck" may be equally or more important. Although competition for carcasses provides a plausible mechanism for generating fitness effects from multigenerational effects, we urge caution in suggesting that this is a strong and pervasive effect. We simply do not know enough about the frequency and intensity of competition for carcasses in natural systems to be confident in asserting the relative importance of body size.

In addition to dominance in competitive environments, larger body size also confers increased resistance to starvation [65]. Resistance to starvation could be important to lifetime fitness in environments where food resources are limited. However, burying beetles can feed on many forms of carrion that are otherwise unsuitable for reproduction, and we know almost nothing about the relative abundance of carcasses for feeding in burying beetle communities, and the corresponding threat of starvation for adult burying beetles. Thus, similar to the argument for the potential selective importance of body size to competition for carcasses, limited resources and the possibility of starvation is likely to be experienced only episodically in burying beetle populations.

In summary, the lack of relationship between multigenerational effects on body size and offspring lifetime fitness suggests that these maternal effects are best understood as "selfish maternal effects" *sensu* [4]. Females appear to be maximizing their own fitness even though that may have no effect on individual offspring lifetime fitness. If, on the other hand, females were using cues from their current environment (such as a large carcass with no competition, or a small carcass with many competitors) in an anticipatory way to determine offspring body size [4], then we might expect natal carcass size to have an effect on lifetime fitness in a similar environment. In our experiment, this should have resulted in a significant two-way interaction between natal carcass size and current carcass size. However, this interaction term was not significant in any of our three measures of lifetime fitness. We reiterate the caution from Marshall and Uller [4] that fitness of offspring resulting from multigenerational effects should not be assumed based on phenotypic traits of offspring, but should be based on experiments that measure lifetime fitness of offspring.

## Acknowledgments

We thank the Charles Redd Center and the Biology Department at Brigham Young University for graduate support for Ashlee Momcilovitch. We thank the numerous Belk Laboratory undergraduate research assistants from 2014 to 2015 for their contributions to maintenance of the beetles in this experiment.

## Author Contributions

**Conceptualization:** Ashlee N. Smith, Mark C. Belk.

**Data curation:** Ethan P. Damron, Dane Jo, Mark C. Belk.

**Formal analysis:** Ashlee N. Smith, Dane Jo, Mark C. Belk.

**Funding acquisition:** Ashlee N. Smith, Mark C. Belk.

**Investigation:** Ethan P. Damron, Ashlee N. Smith, Dane Jo, Mark C. Belk.

**Methodology:** Ashlee N. Smith, Dane Jo, Mark C. Belk.

**Project administration:** Ashlee N. Smith, Mark C. Belk.

**Resources:** Ashlee N. Smith, Mark C. Belk.

**Supervision:** Ashlee N. Smith, Mark C. Belk.

**Visualization:** Ethan P. Damron, Mark C. Belk.

**Writing – original draft:** Ethan P. Damron, Dane Jo.

**Writing – review & editing:** Ashlee N. Smith, Mark C. Belk.

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
