## [Decision Letter · Decision Letter 0]

23 Apr 2021

PONE-D-21-10116

No evidence for increased fitness from multigenerational effects of parental size or natal resource quality in the burying beetle Nicrophorus marginatus

PLOS ONE

Dear Dr. Belk,

Thank you for submitting your manuscript to PLOS ONE. After careful consideration, we feel that it has merit but does not fully meet PLOS ONE’s publication criteria as it currently stands. Therefore, we invite you to submit a revised version of the manuscript that addresses the points raised during the review process.

This manuscript has been assessed by two reviewers who both felt it had a lot of strengths, but also have a number of comments that require further clarification. Please address all of these comments when revising your manuscript.

We look forward to receiving your revised manuscript.

Kind regards,

William David Halliday, Ph.D.

Academic Editor

PLOS ONE

Journal Requirements:

In your Methods section, please provide additional information regarding the permits you obtained for the work. Please ensure you have included the full name of the authority that approved the field site access and, if no permits were required, a brief statement explaining why.

We note that you have stated that you will provide repository information for your data at acceptance. Should your manuscript be accepted for publication, we will hold it until you provide the relevant accession numbers or DOIs necessary to access your data. If you wish to make changes to your Data Availability statement, please describe these changes in your cover letter and we will update your Data Availability statement to reflect the information you provide.

Reviewers' comments:

Reviewer's Responses to Questions

**Comments to the Author**

1. Is the manuscript technically sound, and do the data support the conclusions?

Reviewer #1: Yes

Reviewer #2: Yes

2. Has the statistical analysis been performed appropriately and rigorously? 

Reviewer #1: Yes

Reviewer #2: Yes

3. Have the authors made all data underlying the findings in their manuscript fully available?

Reviewer #1: Yes

Reviewer #2: Yes

4. Is the manuscript presented in an intelligible fashion and written in standard English?

Reviewer #1: Yes

Reviewer #2: Yes

5. Review Comments to the Author

Reviewer #1: The manuscript tests whether variation in two aspects of the parental environment, i.e. parental body size and size of breeding resources, affects offspring fitness. The results suggest that there are no such multigenerational effects. The authors employ a sound experimental manipulation and carefully interpret the results in this very well written manuscript. My main comment is to develop the rationale for the study (see my comment below). When I first read the methods, I was also concerned that the sample size may be too small to have confidence that null results are likely truly null (91 data points across 16 treatment groups). However, the authors found statistically significant effects elsewhere with the same sample size. If trans-generational effects are expected to be of a similar magnitude, there should have been enough power to detect them. Other than that, I only have minor suggestions.

A point that I think is important to emphasise throughout the paper is why and when (adaptive) multigenerational effects are expected to evolve in this system. Burying beetles experience a lot of variation in the size of the carcasses they encounter in the wild. Body size also varies a lot between generations because it is contingent on sibling competition and resource acquisition during larval development. Given so little reliability of the future environment for offspring, it seems unlikely that anticipatory effects could evolve. And, if they exist, these effects might be very small relative to effects of the current environment, such as size of the current carcass. Any such multigenerational effects could thus be masked by the influence of current environment. In other words, it is not clear to me why we should expect to see multigenerational effects in burying beetles. I suggest that you make a stronger case and clarifying the arguments for why this species provides a good system to investigate multigenerational effects.

In some places the authors use "resource quality" to refer to "carcass size", which can be confused with other aspects of the resource (e.g. nutritional value, freshness). This confusion can be avoided by using "amount of resources", or simply "carcass size" or "carcass mass" in place of "quality".

In the abstract, you point out that there was no evidence for increased fitness as a result of multigenerational effects (line 32). Given that these effects could be in both directions, increasing or decreasing fitness, I suggest rephrasing to make the sentence more general. For example, you could say that there was no evidence that multigenerational effects contributed to fitness or, to be more explicit about directionality, that there was no evidence for increased fitness resulting from multigenerational effects in favourable natal environments.

The introduction provides a clear account of the topic and introduces the study system very well. My only comment would be to add predictions regarding the direction of the multigenerational effects. Some of these predictions are already provided in the last paragraph of the discussion. It would be useful to have them earlier on so that the reader is better prepared to assess the methods and results. This could also help motivating the study and provide reasons as to why we should expect multigenerational effects on fitness in burying beetles. Is it because carcass size in previous generations would be a reliable cue for carcass size or the intensity of competition in current generation?

In lines 80-81, do you consider effects of the current carcass as multigenerational effects? This is not necessarily intuitive given that the size of the carcass can impact directly on the offspring feeding upon it, even though parents secure the carcass.

Reviewer #2: Reviewer comments for PLOS ONE manuscript PONE-D-21-10116

This manuscript describes an experimental study examining potential lifetime fitness effects transmitted through parental body size and natal environment quality. The study uses the burying beetle Nicrophorus marginatus which uses the carcasses of dead vertebrates as breeding resource. The authors generated multigenerational effects by varying the size of both parents and the size of the carcass in a parental generation and monitored subsequent effects on the lifetime fitness of female offspring when breeding on carcasses of different size. There was no evidence that parental body size or natal resource size had an effect on lifetime fitness of offspring measured as lifespan, lifetime number of offspring and number of offspring in the first brood.

This study addresses a gap in our understanding of maternal effects by examining whether such effects actually translate into differences in offspring fitness. This study therefore builds constructively upon prior work which has shown consequences for offspring phenotype but in proxies for fitness rather than fitness itself. Overall, I found this to be a clear, well-written and well-motivated study. However, I have some concerns about how these results would play out in the natural ecological context of these beetles. I also have some lingering questions about the statistical analyses and decisions in the experimental design.

Main comments

1. My main concern is to do with the ecological context of the study. As the authors acknowledge in their discussion their experimental design removed competition over carcasses. However, competition is likely to be a very important component of lifetime fitness in these beetles. This is because carcasses are required for reproduction and therefore only beetles that are able to successfully compete and win control of a carcass will be able to produce any offspring at all. Furthermore, body size is the key determinant of success in such competition. In other words – only offspring of sufficient size to be competitive will be capable of securing any reproductive success/fitness at all.

This means that the most important link between maternal effects and offspring fitness is through effects on competitive ability (mediated through body size) rather than the ability to rear offspring in a competition free environment. Thus, by divorcing the study from this natural context the authors may be underestimating the potential importance of these maternal effects to offspring fitness. That is - although the authors show that multigenerational effects do not influence offspring’s lifetime fitness in the absence of competition – it is possible that in a more natural context (that includes competition) such multigenerational effects may be very important.

The authors do acknowledge this point in their discussion (lines: 271-288). However, I think this caveat needs to be worded more clearly and more strongly (not least for the benefit of readers unfamiliar with burying beetle ecology). For example, in their discussion the authors somewhat dismiss the importance of competition by arguing that population density & carcass availability are likely to fluctuate and that therefore competition (and any selection on body size) will be periodical. However, surely this is only true if there are years where carcasses are so abundant that small beetles are able to secure them without competition. Is such a scenario likely? It’s hard to say! This argument also ignores the fact that when suitable carcasses are scarce (which I would imagine is not uncommon even if there is variation) then any selection will be very strong – as only suitably large beetles will be able to gain any reproductive success at all.

It would be nice to see the authors acknowledge more clearly the role of competition in determining offspring fitness in this system.

2. I wonder why the authors chose to analyse their data using parental size as a single factor with four levels (large♀-large♂, large♀-small♂, … etc.). This approach assumes that the effects of parental body size are the same for both sexes and that there are no interactive effects of male and female body size. Given that the authors varied both male and female size separately in a crossed design wouldn’t a more informative approach have been to include male size and female size as separate factors (each with two levels) and also to include the interaction between the two?

3. Can the authors provide justification for the particular range of carcass sizes used in the study? For example, are N. marginatus typically found on both 20 & 40 g carcasses in the wild?

The reason I ask is because although larger carcasses provide more resources they are also potentially harder for the beetles to bury and prepare. Therefore, a carcass that is too large may represent a challenge to reproduction. I assume this is not the case, but it would be reassuring to know that these beetles are not being presented with a challenge they are unlikely to be adapted to.

4. I may be misunderstanding the results here but did the authors find that offspring reared by larger females (or on larger carcasses) eclosed into larger adults in this study? One lines 226-229 they reference finding such an effect from a previous study but it was not clear to me whether the same effect was detected here.

5. In a similar vein some of the data that appeared to be collected is not presented in the results. For example, it would be interesting to see if the proxies for offspring fitness that are sometimes used by other studies (e.g. offspring body mass at dispersal) present a different pattern as the measures for lifetime number of offspring. This might reinforce the authors argument that proxies based on offspring phenotype are not reliable measures of offspring fitness.

6. Could you provide information on the number of reproductive bouts that females in second generation achieved before they died? Based on the number of offspring in the first bout vs lifetime number of offspring it seems like females are breeding 2 or 3 times? Is that correct? It would be useful to know how many times a female is able to breed under these ideal circumstances (well fed, ready supply of mates, ready supply of carcasses and no competition) because as I argued earlier – it seems likely that in burying beetles some individuals will die without ever having secured a carcass to breed which is why potential maternal effects on body size might be more important.

7. Females in the second generation always bred on the same size carcasses throughout their lifespan (either small or large). I understand the rational behind this (as varying carcass size between bouts would make the experiment very unwieldy) but I wonder if the authors found any evidence that consistently breeding on a high or low quality resource influenced reproductive decisions in any way? For example, do females that keep getting small carcasses decide that things are not going to get better and so invest more (i.e. produce more offspring) on their second or third small carcass than they do on the first? This is likely beyond the scope of the current paper but I am curious if the authors have considered this at all.

Minor comments

1. A diagram of the experimental design would be useful. The methods are explained very clearly but a visual representation of the two generation set up might be helpful.

2. Line 281-284: Is this line referring to the fitness of the female parent or the fitness of her adult offspring? The preceding statement seems to be referring to the fitness of offspring – which I think should be the focus of this paragraph – but this line seems to refer to the fitness of the female parent. Can you clarify this?

3. Line 226: Typo – the period after “beetles” should be a comma I think.

4. Line 197-199: It would be helpful to clarify why this particular interaction was included. If I understand correctly, it is because it allows the authors to test if parents “prime” their offspring for reproduction on a particular carcass size – which is one way that multigenerational effects may act. For example, if female offspring reared on a 40g carcass produced more offspring when breeding on a 40g (in comparison to female offspring reared on 20g breeding on 40g).

6. PLOS authors have the option to publish the peer review history of their article (what does this mean?). If published, this will include your full peer review and any attached files.

Reviewer #1: No

Reviewer #2: No

---

## [Author Response · Author response to Decision Letter 0]

7 Jun 2021

Reviewer #1: The manuscript tests whether variation in two aspects of the parental environment, i.e. parental body size and size of breeding resources, affects offspring fitness. The results suggest that there are no such multigenerational effects. The authors employ a sound experimental manipulation and carefully interpret the results in this very well written manuscript. My main comment is to develop the rationale for the study (see my comment below). When I first read the methods, I was also concerned that the sample size may be too small to have confidence that null results are likely truly null (91 data points across 16 treatment groups). However, the authors found statistically significant effects elsewhere with the same sample size. If trans-generational effects are expected to be of a similar magnitude, there should have been enough power to detect them. Other than that, I only have minor suggestions.

A point that I think is important to emphasise throughout the paper is why and when (adaptive) multigenerational effects are expected to evolve in this system. Burying beetles experience a lot of variation in the size of the carcasses they encounter in the wild. Body size also varies a lot between generations because it is contingent on sibling competition and resource acquisition during larval development. Given so little reliability of the future environment for offspring, it seems unlikely that anticipatory effects could evolve. And, if they exist, these effects might be very small relative to effects

of the current environment, such as size of the current carcass. Any such multigenerational effects could thus be masked by the influence of current environment. In other words, it is not clear to me why we should expect to see multigenerational effects in burying beetles. I suggest that you make a stronger case and clarifying the arguments for why this species provides a good system to investigate multigenerational effects.

Our goal in this paper is not to suggest that burying beetles provide a good system in which to find adaptive multigenerational effects. Many multigenerational effects have been documented in burying beetles, including anticipatory effects, and it has been suggested that these already documented effects might translate into increased fitness. We simply wanted to test whether that link to fitness (i.e., larger number of offspring in a lifetime) exists. To make this clear we added the following.

“Clearly, multigenerational effects on offspring size in burying beetles may be transmitted through both parental size and carcass size (i.e., natal environment). The implied prediction is that these multigenerational effects are anticipatory and adaptive [4] such that larger offspring would produce more offspring over their lifetime and thus express increased fitness. However, multigenerational effects may not be adaptive for offspring, rather they can evolve only to increase fitness of the parents [4]. Whether multigenerational effects on body size in burying beetles are adaptive for offspring and translate into increased fitness of offspring is not known. “

We believe this statement clearly lays out our objective and satisfies the reviewer’s comment without

overstating our hypothesis.

In some places the authors use "resource quality" to refer to "carcass size", which can be confused with other aspects of the resource (e.g. nutritional value, freshness). This confusion can be avoided by using "amount of resources", or simply "carcass size" or "carcass mass" in place of "quality".

Done

In the abstract, you point out that there was no evidence for increased fitness as a result of multigenerational effects (line 32). Given that these effects could be in both directions, increasing or decreasing fitness, I suggest rephrasing to make the sentence more general. For example, you could say that there was no evidence that multigenerational effects contributed to fitness or, to be more explicit about directionality, that there was no evidence for increased fitness resulting from multigenerational effects in favourable natal environments.

Good point, we have modified the sentence as follows. “We find no evidence that multigenerational effects from larger parents or larger natal carcasses contribute to increased fitness of offspring.”

The introduction provides a clear account of the topic and introduces the study system very well. My only comment would be to add predictions regarding the direction of the multigenerational effects. Some of these predictions are already provided in the last paragraph of the discussion. It would be useful to have them earlier on so that the reader is better prepared to assess the methods and results. This could also help motivating the study and provide reasons as to why we should expect multigenerational effects on fitness in burying beetles. Is it because carcass size in previous generations would be a reliable cue for carcass size or the intensity of competition in current generation?

We agree that this idea of distinguishing what type of multigenerational effects exist in burying beetles should be introduced in the introduction rather than at the end of the discussion. Accordingly, we have added some text in the introduction that introduces the idea of predicting that these effects could be anticipatory and adaptive for offspring, or they could be adaptive only for the parents following distinctions made in citation number 4 (Marshall and Uller 2007). We have modified our objectives statement to specify that we are testing for adaptive effects that would affect lifetime fitness of offspring. The text now reads:

“Clearly, multigenerational effects on offspring size in burying beetles may be transmitted through both parental size and carcass size (i.e., natal environment). The implied prediction is that these multigenerational effects are anticipatory and adaptive [4] such that larger offspring would produce more offspring over their lifetime and thus express increased fitness. However, multigenerational effects may not be adaptive for offspring, rather they can evolve only to increase fitness of the parents [4]. Whether multigenerational effects on body size in burying beetles are adaptive for offspring and translate into increased fitness of offspring is not known.

In this study we tested whether multigenerational effects of parental size and natal carcass size that result in increased offspring size in the burying beetle Nicrophorus marginatus [65] also cause an increase in lifetime fitness of offspring (i.e., they are adaptive [4]).”

We think this change clarifies our intent in the study.

In lines 80-81, do you consider effects of the current carcass as multigenerational effects? This is not necessarily intuitive given that the size of the carcass can impact directly on the offspring feeding upon it, even though parents secure the carcass.

We do consider this a multigenerational effect and others have suggested that it is a multigenerational effect; however, like many aspects of fitness it can be seen as potentially affecting both parental and offspring fitness. These distinctions are discussed in reference number 4, and are somewhat beyond the scope of the current paper. Although we appreciate the idea, we have not altered our statement because the discussion of this idea is tangential to our main focus.

Reviewer #2: Reviewer comments for PLOS ONE manuscript PONE-D-21-10116

This manuscript describes an experimental study examining potential lifetime fitness effects transmitted through parental body size and natal environment quality. The study uses the burying beetle Nicrophorus marginatus which uses the carcasses of dead vertebrates as breeding resource. The authors generated multigenerational effects by varying the size of both parents and the size of the carcass in a parental generation and monitored subsequent effects on the lifetime fitness of female offspring when breeding on carcasses of different size. There was no evidence that parental body size or natal resource size had an effect on lifetime fitness of offspring measured as lifespan, lifetime number of offspring and number of offspring in the first brood.

This study addresses a gap in our understanding of maternal effects by examining whether such effects actually translate into differences in offspring fitness. This study therefore builds constructively upon prior work which has shown consequences for offspring phenotype but in proxies for fitness rather than fitness itself. Overall, I found this to be a clear, well-written and well-motivated study. However, I have some concerns about how these results would play out in the natural ecological context of these beetles. I also have some lingering questions about the statistical analyses and decisions in the

experimental design. Main comments

1. My main concern is to do with the ecological context of the study. As the authors acknowledge in their discussion their experimental design removed competition over carcasses. However, competition is likely to be a very important component of lifetime fitness in these beetles. This is because carcasses are required for reproduction and therefore only beetles that are able to successfully compete and win control of a carcass will be able to produce any offspring at all. Furthermore, body size is the key determinant of success in such competition. In other words – only offspring of sufficient size to be competitive will be capable of securing any reproductive success/fitness at all.

This means that the most important link between maternal effects and offspring fitness is through effects on competitive ability (mediated through body size) rather than the ability to rear offspring in a competition free environment. Thus, by divorcing the study from this natural context the authors may be underestimating the potential importance of these maternal effects to offspring fitness. That is - although the authors show that multigenerational effects do not influence offspring’s lifetime fitness in the absence of competition – it is possible that in a more natural context (that includes competition) such multigenerational effects may be very important.

The authors do acknowledge this point in their discussion (lines: 271-288). However, I think this caveat needs to be worded more clearly and more strongly (not least for the benefit of readers unfamiliar with burying beetle ecology). For example, in their discussion the authors somewhat dismiss the importance of competition by arguing that population density & carcass availability are likely to fluctuate and that therefore competition (and any selection on body size) will be periodical. However, surely this is only true if there are years where carcasses are so abundant that small beetles are able to secure them

without competition. Is such a scenario likely? It’s hard to say! This argument also ignores the fact that when suitable carcasses are scarce (which I would imagine is not uncommon even if there is variation) then any selection will be very strong – as only suitably large beetles will be able to gain any reproductive success at all.

It would be nice to see the authors acknowledge more clearly the role of competition in determining offspring fitness in this system.

Point well taken! In response we have followed the suggestion of the reviewer and increased our discussion of the potential role of competition for carcasses in conferring fitness. In our expanded discussion of this point we have included three points that we think bear on this idea of strong and pervasive competition as a characteristic of burying beetle biology in the following paragraph.

“ We acknowledge that competitive ability linked to individual body size may confer a fitness advantage under some circumstances, especially if access to carcasses for reproduction is frequently contested, and if body size is often linked to success in obtaining control of carcasses. However, at least three lines of evidence suggest that competition for carcasses may not be a strong determinant of fitness. First, competition for carcasses is dependent on the density of beetles and the availability of small vertebrate carcasses at both large and small scales. Densities of beetles and small vertebrate carcasses fluctuate dramatically among years [74,75]. Similarly, when we trap beetles for this and other studies, we find variation in the number of beetles captured in a given trap, and a range of sizes of both males and females. This spatial variation in number and body size and the temporal variation in beetle numbers and carcass numbers among and within years suggests that competition for

carcasses is likely a local phenomenon. All beetles in a given area are not present and competing for every carcass. Although large beetles may win at carcasses where they compete, some carcasses will have many competitors, and some will have few. Also, when beetles breed on a carcass, they are effectively removed from the potentially competing population for at least two weeks. Thus, the competitive environment is best represented by a mosaic of local carcass availability and local beetle distribution and availability. The outcome is a range of opportunities for beetles of many sizes to be successful at finding and breeding on a carcass. Second, burying beetles exhibit alternative reproductive strategies such that fitness can be obtained by males and females acting as satellites at the carcass burial site or mating prior to finding a carcass [76-78]. If only large individuals can gain access to and reproduce successfully on carcasses, then alternative strategies would not evolve, and body size would be more strongly constrained by selection. Alternative reproductive strategies and a range of body sizes in natural populations suggest that being large is not the only way to be successful in competition for carcasses. Third, in a previous study, we measured competitive ability to gain a carcass based on size in both male and female burying beetles [65]. The relationship differed by sex such that for females, the size difference between competitors had to be much larger for body size to be an important determinant of competitive outcomes. Even for males, there was some non-zero probability of the smaller competitor winning the carcass, except at the largest size difference [65].

Other studies on burying beetles have reported corroborating effects of beetle sizes found on carcasses in natural systems [33]. Body size may be a strong determinant of success in competing for carcasses, but in many contests, priority effects and stochasticity or “luck” may be equally or more important. Although competition for carcasses provides a plausible mechanism for generating fitness effects from multigenerational effects, we urge caution in suggesting that this is a strong and pervasive effect. We simply do not know enough about the frequency and intensity of competition for carcasses in natural systems to be confident in asserting the relative importance of body size.”

We have elaborated our discussion of this idea and included these additional reasons for caution in suggesting that competition for carcasses may be an adaptive reason to expect strong fitness effects of body size.

2. I wonder why the authors chose to analyse their data using parental size as a single factor with four levels (large -large , large -small , … etc.). This approach assumes that the effects of parental body size are the same for both sexes and that there are no interactive effects of male and female body size. Given that the authors varied both male and female size separately in a crossed design wouldn’t a more informative approach have been to include male size and female size as separate factors (each with two levels) and also to include the interaction between the two?

Yes, that would seem obvious, and we appreciate the suggestion because it reminded us that we had not provided sufficient detail in our methods section. We have now included additional information about our choice for the analysis. We have previously analyzed and published the results from the first part (first-generation) of this experiment where we used the factorial design of the parental body size in the way that the reviewer suggested. In that paper, none of the interactions that resulted from treating male and female size as a factorial were significant, so we did not include the interactions in the final analysis of the study. For this study, given the added complexity of including treatments from the second generation, we chose not to analyze the parental size data as a factorial, but rather as 4 combinations. Thus, our highest order interaction is a three-way interaction rather than a four-way interaction and those interactions were tested (they were not significant) and removed (except for one two-way interaction) to provide additional statistical discrimination for main effects. Also,

because there were no significant interactions in the earlier analysis and none in the current analysis, it is unlikely that we would gain any additional insight by keeping parental body size as a factorial effect based on sex. We have added text to explain this choice in analysis as follows.

“We used results from this first-generation experiment to test for effects of body size of parents (male and female separately) and carcass size on their reproductive output and offspring traits [65]. In the first-generation experiment female body size generated significant effects on offspring body size, but male size had no effect on offspring traits and there were no significant interactions between male and female body size [65]. For this reason, we included parental size treatments as four independent treatments in analysis of the second-generation experiment (this paper) rather than as a factorial.

This reduced the complexity of the second-generation analysis by eliminating some two-way and three-way, and all four-way interactions.”

3. Can the authors provide justification for the particular range of carcass sizes used in the study? For example, are N. marginatus typically found on both 20 & 40 g carcasses in the wild?

The reason I ask is because although larger carcasses provide more resources they are also potentially harder for the beetles to bury and prepare. Therefore, a carcass that is too large may represent a challenge to reproduction. I assume this is not the case, but it would be reassuring to know that these beetles are not being presented with a challenge they are unlikely to be adapted to.

Based on a previous study N. marginatus has equal lifetime fitness on a range of carcass sizes from 20 to 40 grams. Thus, both carcass sizes are within the range of carcass sizes they are able to process and utilize. Also, the fact that 40g carcasses yielded higher number of offspring compared to 20g carcasses suggests that beetles were able to fully utilize the larger carcass. We have added text to clarify our choice of carcass size as follows.

“We chose 20g and 40g carcass sizes based on a previous study that tested multiple carcass sizes from 5g to 50 g. The 20g and 40g sizes were both within the range of carcass sizes whereon N. marginatus experienced equally high reproductive success [69].”

4. I may be misunderstanding the results here but did the authors find that offspring reared by larger females (or on larger carcasses) eclosed into larger adults in this study? On lines 226-229 they reference finding such an effect from a previous study but it was not clear to me whether the same effect was detected here.

We have previously published results from the first part (first-generation) of the study reported here (Smith and Belk 2018; citation 65 in the old draft) including that larger females produced larger offsapring. The current analysis is based on the second-generation results. We have now clarified the relationship between these two analyses that represent first-generation and second-generation results as follows.

“We used results from this first-generation experiment to test for effects of body size of parents (male and female separately) and carcass size on reproductive output and offspring traits [65]. In the first- generation experiment female body size generated significant effects on offspring body size, but male size had no effect on offspring traits and there were no significant interactions between male and female body size [65]. For this reason, we included parental size treatments as four independent treatments in analysis of the second-generation experiment (this paper) rather than as a factorial.

This reduced complexity of the second-generation analysis by eliminating some two-way and three- way, and all four-way interactions.”

In the current study (second-generation) we did not test for this effect (i.e., larger females produce larger offspring) because it had already been established in the first generation and it was not our purpose here to inspect phenotype of the offspring, but rather fitness (number of offspring) of the second-generation females.

5. In a similar vein some of the data that appeared to be collected is not presented in the results. For example, it would be interesting to see if the proxies for offspring fitness that are sometimes used by other studies (e.g. offspring body mass at dispersal) present a different pattern as the measures for lifetime number of offspring. This might reinforce the authors argument that proxies based on offspring phenotype are not reliable measures of offspring fitness.

Once again, results of phenotype of offspring were reported in the previous paper as described above (Smith and Belk 2018). In the current analysis we only reported fitness effects. However, the effect of pronotum width of the offspring was included in each analysis, and it was only significant in the analysis of lifespan. We discuss this effect in the first paragraph of the discussion, and show that the added lifetime for larger individuals does not equate to increased fitness. Thus, we have indicated that the proxy of body size is not a reliable measure of fitness.

6. Could you provide information on the number of reproductive bouts that females in second generation achieved before they died? Based on the number of offspring in the first bout vs lifetime number of offspring it seems like females are breeding 2 or 3 times? Is that correct? It would be useful to know how many times a female is able to breed under these ideal circumstances (well fed, ready supply of mates, ready supply of carcasses and no competition) because as I argued earlier – it seems likely that in burying beetles some individuals will die without ever having secured a carcass to breed which is why potential maternal effects on body size might be more important.

Number of reproductive bouts for females in the second-generation experiment ranged from 1 to 5 with a mean of 2.9, and we have included this information in the methods section that explains the second-generation experiment. We agree that in natural systems it is unlikely that females are able to breed as the dominant female on a carcass more than 1 or 2 times. For this reason, we included the number of offspring in the first bout as a measure of fitness. In response to the first question above we have provided several reasons why competitive ability arising from differences in body size may not always be a reasonable expectation for generating increased fitness. We added the following text to the discussion.

“In natural environments, number of successful reproductive bouts may be fewer than that in our

non-competitive lab environment. For this reason, we included number of offspring in the first bout as a potentially more natural alternative to measure lifetime reproductive success. Patterns were the same whether we measured fitness as lifetime number of offspring or number of offspring in the first bout, suggesting that even if females reproduce on only one carcass during their lifetime, there is still no effect of parental body size or natal carcass size.”

7. Females in the second generation always bred on the same size carcasses throughout their lifespan (either small or large). I understand the rational behind this (as varying carcass size between bouts would make the experiment very unwieldy) but I wonder if the authors found any evidence that

consistently breeding on a high or low quality resource influenced reproductive decisions in any way? For example, do females that keep getting small carcasses decide that things are not going to get better and so invest more (i.e. produce more offspring) on their second or third small carcass than they do on the first? This is likely beyond the scope of the current paper but I am curious if the authors have considered this at all.

This is an interesting question, but as noted by the reviewer it is not within the scope of the current paper. We have addressed this effect in a previous paper, and there is an effect of the size of carcass first obtained on subsequent reproductive decisions. See:

Billman, E, CJ, Creighton, MC Belk. 2014. Prior experience affects allocation to current reproduction in a burying beetle. Behavioral Ecology, 25: 813-818.

Minor comments

1. A diagram of the experimental design would be useful. The methods are explained very clearly but a visual representation of the two generation set up might be helpful.

We have considered this suggestion, but we feel that a diagram might be redundant with the already long explanatory text. If the editor feels a diagram is warranted, we would be happy to create one.

2. Line 281-284: Is this line referring to the fitness of the female parent or the fitness of her adult offspring? The preceding statement seems to be referring to the fitness of offspring – which I think should be the focus of this paragraph – but this line seems to refer to the fitness of the female parent. Can you clarify this?

We clarified the statement to refer to fitness of offspring as follows: “Thus, large body size of a female beetle may translate into increased lifetime fitness in her offspring in a natural environment because larger female offspring are able to win access to more and larger carcasses, and thus produce more offspring.”

3. Line 226: Typo – the period after “beetles” should be a comma I think.

Done

4. Line 197-199: It would be helpful to clarify why this particular interaction was included. If I

understand correctly, it is because it allows the authors to test if parents “prime” their offspring for reproduction on a particular carcass size – which is one way that multigenerational effects may act. For example, if female offspring reared on a 40g carcass produced more offspring when breeding on a 40g (in comparison to female offspring reared on 20g breeding on 40g).

Thank you for this suggestion. That idea captures our intent better than we had previously. We have modified the sentence to read as follows: “This interaction was of interest because of the significant effect of current carcass size on fitness measures, and the possibility of a multigenerational effect that natal carcass size might provide some “priming” for efficiency of use of a similar carcass size.

---

## [Decision Letter · Decision Letter 1]

15 Jun 2021

No evidence for increased fitness of offspring from multigenerational effects of parental size or natal carcass size in the burying beetle Nicrophorus marginatus

PONE-D-21-10116R1

Dear Dr. Belk,

We’re pleased to inform you that your manuscript has been judged scientifically suitable for publication and will be formally accepted for publication once it meets all outstanding technical requirements.

Kind regards,

William David Halliday, Ph.D.

Academic Editor

PLOS ONE

Additional Editor Comments (optional):

Reviewers' comments:

Reviewer's Responses to Questions

**Comments to the Author**

1. If the authors have adequately addressed your comments raised in a previous round of review and you feel that this manuscript is now acceptable for publication, you may indicate that here to bypass the “Comments to the Author” section, enter your conflict of interest statement in the “Confidential to Editor” section, and submit your "Accept" recommendation.

Reviewer #1: All comments have been addressed

Reviewer #2: All comments have been addressed

2. Is the manuscript technically sound, and do the data support the conclusions?

Reviewer #1: Yes

Reviewer #2: Yes

3. Has the statistical analysis been performed appropriately and rigorously? 

Reviewer #1: Yes

Reviewer #2: Yes

4. Have the authors made all data underlying the findings in their manuscript fully available?

Reviewer #1: Yes

Reviewer #2: Yes

5. Is the manuscript presented in an intelligible fashion and written in standard English?

Reviewer #1: Yes

Reviewer #2: Yes

6. Review Comments to the Author

Reviewer #1: The authors have responded to all the concerns I raised during the first round of revisions and I have no additional comments. I think that this paper will be an excellent addition to the field and I look forward to its publication.

Reviewer #2: (No Response)

7. PLOS authors have the option to publish the peer review history of their article (what does this mean?). If published, this will include your full peer review and any attached files.

Reviewer #1: No

Reviewer #2: No

---

## [Editor Report · Acceptance letter]

28 Jun 2021

PONE-D-21-10116R1 

No evidence for increased fitness of offspring from multigenerational effects of parental size or natal carcass size in the burying beetle *Nicrophorus marginatus*

Dear Dr. Belk:

I'm pleased to inform you that your manuscript has been deemed suitable for publication in PLOS ONE. Congratulations! Your manuscript is now with our production department. 

Kind regards, 

on behalf of

Dr. William David Halliday 

Academic Editor

PLOS ONE